# Grain Structure Formation and Texture Modification through Multi-Pass Friction Stir Processing in AlSi10Mg Alloy Produced by Laser Powder Bed Fusion

**DOI:** 10.3390/ma16030944

**Published:** 2023-01-19

**Authors:** Akbar Heidarzadeh, Mousa Javidani, Mohammadreza Mofarrehi, Pouyan Motalleb-nejad, Roghayeh Mohammadzadeh, Hamidreza Jafarian, X.-Grant Chen

**Affiliations:** 1Department of Materials Engineering, Azarbaijan Shahid Madani University, Tabriz P.O. Box 53714-161, Iran; 2Department of Applied Science, University of Quebec at Chicoutimi, Saguenay, QC G7H 2B1, Canada; 3Faculty of Materials Engineering, Sahand University of Technology, Tabriz P.O. Box 51335-1996, Iran; 4School of Metallurgy and Materials Engineering, Iran University of Science and Technology, Narmak, Tehran P.O. Box 16846-13114, Iran

**Keywords:** AlSi10Mg alloy, friction stir processing, grain boundary, texture, dynamic recrystallization

## Abstract

A new strategy is proposed to modify the grain structure and crystallographic texture of laser-powder bed fusion AlSi10Mg alloy using multi-pass friction stir processing (FSP). Accordingly, 1–3 passes of FSP with 100% overlap were performed. Scanning electron microscopy and electron backscattered diffraction were used for microstructural characterization. Continuous dynamic recrystallization and geometric dynamic recrystallization are the governing mechanisms of grain refinement during FSP. The stir zones have bimodal grain structures containing large and fine grains. The multi-pass FSP caused a considerable increase in the volume fraction of the large-grained area in the stir zone, which contained higher values of low-angle boundaries and sharp shear texture components of $B\left(1\bar{1}2\right)\left[110\right]$ and $\bar{B}\left(\bar{1}1\bar{2}\right)\left[\bar{1}\bar{1}0\right]$. The formation of low-energy grain boundaries in the stir zone and alignment of the low-energy crystallographic planes with the surface of the sample made the strategy of using multi-pass FSP a promising candidate for corrosion resistance enhancement in future studies. Moreover, the detailed evolution of the grains, texture components, grain boundaries, and Si particles is discussed.

## 1. Introduction

Aluminum alloys are important materials in the aerospace and automotive industries and are appropriate candidates for production using laser-based additive manufacturing owing to their good printability [1]. Among aluminum alloys, AlSi10Mg has attracted significant attention owing to its high strength/weight ratio, low cost, and excellent printability [2]. Currently, the AlSi10Mg alloy is produced and employed in different industrial sectors by laser-powder bed fusion (L-PBF), which is a laser-based AM method [3]. During the last decade, a number of investigations have been undertaken to elucidate the microstructure and mechanical properties of AlSi10Mg, which show the negative effect of L-PBF on ductility and fatigue resistance [4]. Therefore, some researchers have attempted to overcome this problem using post-treatments, including both heat and deformation treatments [5,6,7,8].

Improving the ductility of AlSi10Mg by post-treatment has often been found in conjunction with a decrease in strength [1]. One of the most promising methods for post-treatment of AlSi10Mg is friction stir processing (FSP) [9]. During FSP, which was developed based on the friction stir welding (FSW) concept, the heat and severe plastic deformation induced by a rotational tool cause microstructural changes, such as grain refinement and the fragmentation of secondary phases [10,11]. Thus, FSP can be used to modify the microstructure and mechanical properties of metals and alloys [10]. In the case of L-PBF AlSi10Mg, FSP causes less reduction in strength when ductility is improved [1]. For instance, Macías et al. [9] reported that FSP can increase the ductility and fatigue life time by 448% and 200%, respectively, with only a reduction in yield strength by 34%.

In ref. [9,12,13,14,15,16,17,18,19,20,21], most microstructural studies of friction-stir-processed (FSPed) AlSi10Mg produced by L-PBF have focused on defect elimination, the evolution of the Si structure, and precipitation phenomena. However, fundamental information regarding the grain structure formation during FSP and the corresponding grain boundaries and crystallographic texture components is lacking, which is necessary to be understood to control the final properties of the processed materials [10]. On the other hand, the parameters of FSW and FSP, such as tool geometry, pin to shoulder ratio, shoulder surface features, tool eccentricity, tool rotational speed, tool traverse speed, number of passes, etc., influence the microstructure and mechanical performance [11,12,13,14]. For example, Hou et al. [11] investigated the role of pin eccentricity on microstructural evolution and mechanical properties of FSWed Al-Mg-Si alloy sheets. They found that pin eccentricity had a positive effect on promoting the material flow and grain refinement in the stir zone. The texture intensity was increased as the pin eccentricity was introduced, and the dominant texture component converted from C{001}⟨[110]⟩ (without eccentricity) to $B/\bar{B}\left\{112\right\}\left\langle [110]\right\rangle$ when eccentric pin was used. Senthilkumar et al. [14] studied the effect of the number of passes in FSP of AA6082 alloy. They revealed that increasing the number of passes with the specific rotational speed, the tunnel void at stir zone could be minimized due to the recrystallization mechanism. In addition, it increased the grain size with more dissolution and re-precipitation along with intense fragmentation of second-phase particles. Therefore, in this study, the microstructural evolution in FSPed zones in single-pass and multi-pass AlSi10Mg was studied in detail. The grain structure formation and changes in the eutectic Si structure and texture were characterized using scanning electron microscopy and electron backscattered diffraction (EBSD) to reveal the governing microstructural mechanisms during FSP. Consequently, a strategy has been proposed for grain structure formation and the modification of crystallographic textures in L-PBF AlSi10Mg to enhance the performance of the final products.

## 2. Materials and Methods

AlSi10Mg plates with dimensions of 80 mm length × 40 mm width × 2 mm thickness were produced by the L-PBF method. Gas-atomized AlSi10Mg powders with an average diameter smaller than 65 µm were used in this study. A Noura M100P printer with the following process conditions was employed to fabricate the samples: powder consumption of 80 g, layer thickness of 30 µm, energy density of 60 J/mm^3^, hatch space of 0.19 mm, scan rotation of 67°, number of slices of 260, and overall build time of 2 h. Subsequently, three different friction stir post-processing conditions were employed on the AlSi10Mg L-PBF plates with single, double, and multi-pass FSP passes. To achieve this, the FSP was performed perpendicular to the building direction of the plate at a traverse speed of 85 mm/min, a rotational speed of 950 rpm, a plunge depth of 0.1 mm, and a tilt angle of 2° using a tool made of H13 tool steel, employing a conventional milling machine (Vertical WMW Heckert FSS 315 V/2). It is worth noting that the plunge depth was applied once the tool head touched the surface of the plates. The FSP tool is composed of a pin (with a diameter of 3 mm and a length of 1.7 mm) and a shoulder with a diameter of 12 mm. Single-, double-, and multi-pass FSP were conducted with a 100% overlap to produce different samples denoted here as 1-pass, 2-pass, and 3-pass samples, respectively.

Metallographic samples were cut from the processed samples perpendicular to the FSP direction and prepared using a standard metallographic procedure. The optical macrostructure of the joint was etched with a solution of 5 cc HNO_3_, 3 cc HCl, 2 cc HF, and 190 cc distilled water. The EBSD analysis was performed using a scanning electron microscope (SEM) with a step size of 0.2 µm, and all the corresponding data were processed using TSL-OIM software with a threshold of 15° between low-angle grain boundaries (LABs) and high-angle grain boundaries (HABs). Brandon’s criterion was used for coincidence site lattice (CSL) boundary classification in TSL-OIM software. Moreover, SEM was employed to observe the different structures of the Si-rich phase in the L-PBF as-built and FSP-treated states.

## 3. Results

### 3.1. Grain Structure Formation

The inverse pole figure (IPF) and image quality (IQ) maps of the as-built L-PBF AlSi10Mg, which are treated here as the base material (BM) of the FSP, are presented in Figure 1. The BM is composed of two types of grains: large elongated grains (black arrows in Figure 1a) and fine equiaxed grains (white arrows in Figure 1a) within the melt pools, as indicated by the red arrows in Figure 1b. The average grain size of BM was 16.1 µm. The elongated grains are primarily formed parallel to the BD, which is against the heat transfer direction during the solidification of melt pools [1]. The formation of fine equiaxed grains occurs when MP is consumed by elongated grains, and the temperature gradient decreases to a certain value for the nucleation of equiaxed grains [22]. From Figure 1, the BM contains 78 and 22% HABs and LABs, respectively. It is well documented that FSP destroys the initial microstructure of BMs owing to various restoration mechanisms [10,15]. However, in the case of L-PBF AlSi10Mg, the grain structure formation, restoration mechanisms, grain boundaries, and texture evolution during FSP have not yet been disclosed, which is discussed in detail in the following sections.

#### 3.1.1. Single Pass FSP

The cross-sectional macrostructure of the 1-pass sample is shown in Figure 2a, indicating the presence of distinct zones including the BM, transition zone, and stir zone (SZ). The BM was not affected by heat or deformation during FSP, exhibiting a typical microstructure of L-PBF AlSi10Mg (Figure 1) containing overlapping melt pool boundaries. However, the transition zone between the BM and SZ (Figure 2b), which is usually called the thermomechanically affected zone (TMAZ), is affected during FSP. Upon reaching the center of the SZ, the temperature, strain, and strain rate induced by the rotational tool increased to their maximum values. Hence, microstructural evolution can be completed in the SZ, whereas it occurs partially in the transition zone owing to inadequate heat and deformation. As shown in Figure 2b, the SZ was divided into two regions, SZ-I and SZ-II, which have different dark and bright contrasts. The four blue rectangles in Figure 2b are the different areas of the EBSD scans from TMAZ (rectangle 1) to SZ-I (rectangle 4). It is notable that the characterization of the TMAZ can be used to reveal the microstructural evolution during grain structure formation because it is the transition zone including incomplete mechanisms between the BM and SZ [23,24].

The inverse pole figure map of rectangle 1 in Figure 2b is shown in Figure 3. The higher magnified views of the three different zones in conjunction with the grain boundaries (HABs: black color and LABs: white color) are illustrated in Figure 3b–d. There was a gradient of strain, temperature, and strain rate from the BM to the center of the SZ during the FSP [15], which caused the formation of different microstructural zones, as shown in Figure 3. At low strain and temperature (Figure 3b), the grains were elongated owing to the shear deformation induced by FSP, and the melt pool boundaries of the BM disappeared. The presence of subgrains surrounded by LABs are confirmed by black arrows in Figure 3b. Evidently, the grain structure formation can be started with a mechanism accelerated by subgrains. In the zones with higher temperature and strain during FSW (Figure 3c,d), the LABs were gradually transformed to HABs, as indicated by the blue arrows, which confirms the occurrence of continuous dynamic recrystallization (CDRX). Upon increasing the strain value, more dislocations were absorbed into the LABs, resulting in a gradual increase in their misorientations. When the misorientation of the LABs is larger than 15° (the threshold for identifying HABs), they transform into HABs; hence, new DRX grains are formed. In Figure 3d, in addition to CDRX, geometric dynamic recrystallization (GDRX) was detected in the TMAZ, as indicated by red arrows. The higher temperature and strain values in this zone compared to those in (Figure 3b,c) caused the formation of more elongated/fibrous grains. When the width of the elongated grains was approximately equal to the grain size of the SZ, the serrations of the old grain boundaries touched each other, and GDRX grains could be formed [25]. Notably, GDRX is usually categorized as a type of CDRX. Thus, it can be concluded that at the initial stage of hot deformation in the 1-pass sample during FSP, the grain structure formation begins with DRV, followed by CDRX through LABs→HABs transformation and GDRX. The other point in Figure 3 is the heterogeneous structure within the TMAZ, which indicates non-uniform plastic deformation during FSP causing different amounts of dynamic recrystallization in different zones (Figure 3c,d).

The inverse pole figure map of rectangle 2 in Figure 2b is shown in Figure 4. The enlarged views of zones 1 and 2 in Figure 4a in conjunction with the HABs and LABs are illustrated in Figure 4b,c. From Figure 4, at higher levels of strain and temperature during FSP, more DRX grains were generated within the TMAZ because the DRV and CDRX mechanisms can occur in higher quantities compared to the areas with lower strain and temperature (Figure 3). The inverse pole figure map of the rectangle 3 in Figure 2b are shown in Figure 5. A further increase in strain and temperature in the TMAZ, that is, by approaching the TMAZ and SZ-II interface in Figure 2b, the DRX was completed, as shown in Figure 5b. As shown in Figure 5c, SZ-II contained larger grains than the TMAZ, which is attributed to the higher temperature in this area during FSP, which encouraged grain growth. The origin of the different regions in the SZ in the cross-sectional macrostructure (Figure 2), that is, SZ-I and SZ-II, is shown in the inverse pole figure maps in Figure 6, which illustrates the interface between these zones. SZ-II (Figure 6a) was composed of larger grains than SZ-I (Figure 6b), resulting in a bimodal grain structure in the SZ. This effect can be explained by the correlation between the thermomechanical behavior of the material using the Zener–Holloman parameter (*Z*) [26]:
(1)$$Z=\frac{d\varepsilon }{dt}exp\left(\frac{Q_{def}}{RT}\right)$$
where $\frac{d\varepsilon }{dt}$ is the strain rate, $Q_{def}$ is the activation energy required for the deformation, *R* is the gas constant, and *T* is the deformation temperature. The *Z* value indicates the grain size of metals and alloys that undergo plastic deformation during thermomechanical processing. The relationship between *Z* value and average grain size ($d_{Rx}$) can be expressed as follows [27,28]:(2)$$Zd_{Rx}{}^{m}=A$$
where *A*, $d_{Rx}$, and *m* are the constant, DRX grain size, and grain-size exponent, respectively. Equations (1) and (2) reveal that the strain rate and temperature are critical in determining the final grain size of FSPed metals and alloys, which have opposite effect on final grain size ($d_{Rx}$). From Figure 6 and considering Equations (1) and (2), the strain rate ($\frac{d\varepsilon }{dt}$) and temperature (T) are not the same in SZ-I and SZ-II. The finer grain size in SZ-I (Figure 6c) is related to lower temperature and higher strain rate compared to those in SZ-II. In contrast, the larger grain size in SZ-II (Figure 6b) is attributed to the higher temperature and lower the strain rate in this area. From literature [10,15], the complex material flow during FSW/P causes the formation of distinct zones with different local deformation and temperature values, and hence different microstructures such as onion rings. Similarly, as mentioned before, the existence of two distinct zones (SZ-I and SZ-II) with different grain size and sharpness can be explained by qualitative analysis using Equations (1) and (2). Therefore, a combination of temperature and strain rate can be applied to achieve the grain size revealed by the EBSD measurements in SZ-I and SZ-II. For this aim, combined simulation/experimental research is needed to measure the exact values of strain, strain rate and temperature in different zones during FSW/P, which can be a topic for future study.

#### 3.1.2. Double and Multiple Pass FSP

The cross-sectional macrostructure of 2-pass FSPed samples is shown in Figure 7, which indicates a similar behavior to the 1-pass FSPed sample (Figure 2). The fractions of SZ-II and SZ-I did not change considerably after the second pass of the FSP. The EBSD scan areas are shown by blue rectangles in Figure 7. The inverse pole figure maps corresponding to the TMAZ are shown in Figure 8. As shown in Figure 8b,c, the second pass of FSP caused a fully equiaxed grain structure in the TMAZ. During the second pass of the FSP, the non-DRX parts of the TMAZ after the single-pass FSP (Figure 3 and Figure 4) underwent CDRX, and hence the DRX was completed in this area. When approaching the SZ-II region in Figure 7, coarser grains appeared, as shown in the inverse pole figure maps in Figure 9. Therefore, according to Figure 7, Figure 8 and Figure 9, the second pass of the FSP completes the partial restoration mechanisms during the single-pass FSP. The interfacial area between SZ-II and SZ-I is similar to that of the 1-pass samples, and for the sake of brevity, the corresponding inverse pole figures are not presented.

The cross-sectional macrostructure of the 3-pass sample is presented in Figure 10, indicating the presence of different microstructural zones similar to those of the 1- and 2-pass samples (Figure 2 and Figure 7, respectively). However, unlike the 2-pass sample (Figure 7), the third pass of FSP resulted in a considerable increase in the volume fraction of the SZ-II region.

The inverse pole figure maps of the interfacial area between the TMAZ and SZ-II (indicated by rectangle 1 in Figure 10) are presented in Figure 11. From the perspective of grain structure, the TMAZ and SZ-II in the 3-pass sample had similar characteristics to the 2-pass sample. The existence of considerable LABs (Figure 11) confirmed that the grains underwent deformation after their formation by DRX. The inverse pole figure map of the interfacial area between SZ-II and SZ-I (area indicated by rectangle 2 in Figure 10) is shown in Figure 12a. The higher magnification inverse pole figure maps of SZI and SZ-II are shown in Figure 12b,c, respectively. From Figure 12a–c, SZ-I contained finer grains compared to those of SZ-II, similar to those of the 1- and 2-pass samples. Similar to the grain structure, SZ-I and SZ-II exhibit strong texture formation with a bimodal nature. The texture components formed in different samples are discussed in detail in the following sections.

### 3.2. Evolution of Si Structure

The SEM images of BM, SZ-I, and SZ-II for different samples perpendicular to the BD are illustrated in Figure 13 to reveal the evolution of the Si structure during the FSP. As shown in Figure 13(a1), the melt pools were composed of three distinct zones: coarse melt pool (Figure 13(a2)), heat-affected zone (Figure 13(a3)), and fine melt pool (Figure 13(a3)), which is in agreement with those reported in the literature [1]. As shown in Figure 13b,c, FSP caused fragmentation of eutectic Si cells into Si particles. In addition, by increasing the number of FSP passes, a slight increase in Si particle size was detected. Moreover, in all cases, SZ-II (Figure 13(b1–d1)) contained larger Si particles than SZ-I (Figure 13(b2–d2)) did. During FSP, which induces heat into the material, the Si particles coalesce or join to the surface of the larger Si particles, resulting in the continuous growth of Si particles. The areas indicated by the blue arrows in Figure 13(c1–c3) confirm the growth mechanism of the Si particles. Moreover, the presence of Si particles at the grain boundaries (indicated by the yellow arrow in Figure 3b) showed the Zener pinning effect [25], which has a negative effect on the mobility of grain boundaries, and hence, the grain growth mechanism. The presence of large Si particles in SZ-II, with diameters larger than 1 µm, can be a suitable condition for the occurrence of particle stimulated nucleation [25]. For example, in the case of the 3-pass sample, fine grains were observed around the non-indexed zones in the inverse pole figure maps of SZ-II, as shown by the circles in Figure 11c, which can be an indication of the particle-stimulated nucleation mechanism.

## 4. Discussion

The grain size, grain boundary characterization, and coincidence site lattice boundary distributions in both the SZ-I and SZ-II areas in all the samples are illustrated in Figure 14a–c. From the grain size distribution (Figure 14(a1,a2)), FSP results in the formation of a fine grain structure with an average grain size ($d_{av}$) between 2.5 and 3.7 µm. Grain refinement is due to the occurrence of CDRX and GDRX.

Moreover, in the case of SZ-I, the second pass of FSP caused an increase in $d_{av}$ from 2.6 µm to 3.1 µm (Figure 14(a1)) and an enhancement in LABs from 22% to 32% (Figure 14(b1)), which indicated the occurrence of grain growth and formation of LABs through DRV. However, the third pass of FSP consumed the LABs and transformed them to HABs (Figure 14(b1)), which reduced the $d_{av}$ from 3.1 µm to 2.5 µm (Figure 14(a1)). In the case of SZ-II, there was an opposite way compared to SZ-I, in which the second pass of FSP caused a reduction in $d_{av}$ from 3.7 µm to 3.2 µm (Figure 14(a2)) by reducing the amount of LABs from 26% to 20% (Figure 14(b2)). Furthermore, the third pass resulted in an increase in the amount of LABs (34%) and $d_{av}$ of 3.5 µm (Figure 14(a2,b2)). From the perspective of $d_{av}$, single- pass and multi-pass FSP resulted in a bimodal grain structure in SZ and finer grains ($d_{av}$ = 2.6 µm and 2.5 µm) in SZ-I and coarser grains ($d_{av}$ = 3.7 µm and 3.5 µm) in SZ-II, respectively. However, in the case of the 2-pass sample, there was no considerable difference between the $d_{av}$ values of SZ-I and SZ-II. However, from the viewpoint of the Si phase, in all samples, there was a bimodal distribution of finer (SZ-I) and coarser Si particles (SZ-II) in the SZs (Figure 13). Therefore, the different contrasts of SZ-I and SZ-II in the cross-sectional macrostructures (Figure 2, Figure 7 and Figure 10) were due to the different grain and Si particle sizes of these zones. From Figure 14b, FSP has not a considerable effect on the amount of coincidence site lattice boundaries compared to that of the BM. The largest change in the number of low energy coincidence site lattice boundaries, i.e., Σ ≤ 27, belongs to the SZ-I in 3-pass sample containing 10.1% coincidence site lattice boundaries, which indicates only a 2% enhancement. Moreover, the Σ3 boundaries have not a considerable fraction in coincidence site lattice boundaries ranged between 0.96% and 2.54%.

The inverse pole figures parallel to the x, y, and BD directions, in conjunction with the (001), (011), and (111) pole figures (PFs) of the BM, are illustrated in Figure 15a,b, respectively. As shown in Figure 15a,b, a sharp texture of BD//[001] is generated because the preferred solidification direction in face-centered cubic (FCC) metals, that is, the [001] crystallographic direction, aligns along the heat transfer direction, which is almost parallel to BD [22].

Similar to the grain structure, FSP destroys the crystallographic texture of metals and alloys owing to severe plastic deformation induced by the rotational tool [29]. The deformation of metals and alloys during FSP is shear nature [29]. Therefore, to analyze the texture components of SZs, they are usually compared to those of simple shear deformation, as summarized in Table 1 for FCC materials [10]. It should be noted that the as-acquired EBSD data should be rotated to adjust it to the standard reference frame of a simple shear illustration [10].

The rotated (001), (011), and (111) pole figures of the SZ-I and SZ-II areas in the 1–3 pass FSPed samples are illustrated in Figure 16. In the case of SZ-I, the first pass of FSP formed the shear texture component of $A_{1}^{*}\left(111\right)\left[\bar{1}\bar{1}2\right]$ (Figure 16(a1)), whereas the second and third passes resulted in the formation of $A_{2}^{*}\left(111\right)\left[11\bar{2}\right]$ component in addition to the $A_{1}^{*}\left(111\right)\left[\bar{1}\bar{1}2\right]$ component (Figure 16(b1,c1)). The formation of the $A_{1}^{*}\left(111\right)\left[\bar{1}\bar{1}2\right]$ component might be due to more shear deformation induced by the FSP during the second and third passes, which could activate more crystallographic slip systems. In the case of SZ-II, $A_{1}^{*}\left(111\right)\left[\bar{1}\bar{1}2\right]$ and $B\left(1\bar{1}2\right)\left[110\right]$ were the main shear texture components in the 1-pass sample (Figure 16(a2)). The second pass of FSP (Figure 16(b2)) generated trimodal texture components in SZ-II composed of $A_{1}^{*}\left(111\right)\left[\bar{1}\bar{1}2\right]$, $B\left(1\bar{1}2\right)\left[110\right]$, and $\bar{B}\left(\bar{1}1\bar{2}\right)\left[\bar{1}\bar{1}0\right]$, indicating a reduction in the fraction of $A_{1}^{*}$ in the total crystallographic texture by the onset of $\bar{B}$ component. From Figure 16(c2), interestingly, the multipass FSP caused the complete elimination of $A_{1}^{*}$ by the generation of $B\left(1\bar{1}2\right)\left[110\right]$ and $\bar{B}\left(\bar{1}1\bar{2}\right)\left[\bar{1}\bar{1}0\right]$ components with a considerable texture intensity of 12.9. It is worth noting that the formation of sharp shear texture components in SZs (Figure 16) confirms the suggested mechanisms, as CDRX and GDRX inherited the texture components induced by shear deformation during FSP [10]. The inverse pole figure maps of SZ-I and SZ-II after the applied rotations for texture analysis are shown in Figure 17a,b, respectively. In SZ-II, the dense (111) planes were approximately parallel to the surface of the sample (Figure 17b), whereas in SZ-I, the dense planes were not aligned with the surface (Figure 17a).

The multi-pass FSP increased the fraction of SZ-II containing fine grains, coarse Si particles, a large number of LABs, and $B/\bar{B}$ shear texture components. These findings indicate that by controlling the number of passes during the FSP of an L-PBF AlSi10Mg alloy, the grain structure and crystallographic texture can be modified in conjunction with the Si structure. This strategy can be a platform for future investigations to enhance the performance of L-PBF AlSi10Mg in various sectors, according to the corresponding conditions of industrial applications. For instance, it is well documented that aluminum alloys containing more low-energy grain boundaries (LABs) and coincidence site lattice boundaries (Σ ≤ 27, especially Σ3) exhibit higher corrosion resistance [31]. Moreover, crystallographic planes with lower surface energies, such as the (111) plane, have higher corrosion resistance [32,33,34]. Efforts to disclose the effect of FSP on the electrochemical response, as an important final property, of L-PBF AlSi10Mg are very limited. Rafieazad et al. [21] employed a single-pass FSP on an L-PBF AlSi10Mg alloy and studied the corrosion behavior of the as-built and processed samples. They reported that a single-pass FSP improved the corrosion resistance by eliminating the porosity, grain refinement, and uniform distribution of Si particles in the SZ. The grain structure and texture characteristics obtained in this study can be used to further improve corrosion resistance, which requires further investigation in the future. This strategy can also be applied to modify the physical and mechanical responses of the L-PBF Al-Si10Mg alloy according to the requirements of industrial applications.

## 5. Conclusions

Single-, double-, and multi-pass FSPs were used to modify the microstructure and texture of the L-PBF AlSi10Mg alloy. The following conclusions can be drawn:
(1)During the first pass of FSP, dynamic recovery, continuous dynamic recrystallization and geometric dynamic recrystallization caused the formation of a fine grain structure in the SZ with an average grain size of 2.6–3.7 µm. The bimodal grain structure results from the formation of two distinct zones: SZ-I with a fine grain size and SZ-II with a coarse grain size in the SZ.(2)The second pass of FSP completed the recrystallization in the TMAZ. However, in the case of SZ, it causes the rotation of grains, which generates more shear texture components in SZ-I and SZ-II compared with single-pass FSP.(3)The third pass of FSP increased the volume fraction of SZ-II, which possessed unique characteristics of higher values of LABs. During the third pass, more rotation of DRX grains in SZ occurred, which resulted in the generation of $B\left(1\bar{1}2\right)\left[110\right]$ and $\bar{B}\left(\bar{1}1\bar{2}\right)\left[\bar{1}\bar{1}0\right]$ components with a considerable texture intensity of 12.9 in SZ-II. The produced texture components in SZ-II aligned the crystallographic planes with lower surface energy, such as {111} planes parallel to the surface of samples.(4)In addition, in all cases, the Si morphology changed from cell to particle structure in SZs. Similar to grain size distribution, the SZ-II contained larger Si particles compared to the SZ-I.(5)The outcome of this study can be used for the modification of the microstructure (grain size, grain boundaries, and texture) of L-PBF AlSi10Mg alloy in a manner that, by controlling the number of passes during FSP, the desired microstructural features can be obtained for potential applications.


## Figures and Tables

**Figure 1 materials-16-00944-f001:**
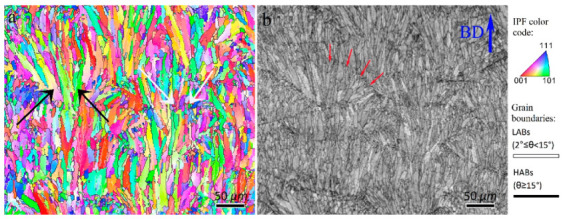
(**a**) Inverse pole figure and (**b**) image quality maps of as-built L-PBF AlSi10Mg. High and low angle grain boundaries are drawn in black and white colors on all inverse pole figure maps through paper, respectively. Black and white in (**a**) arrows show large-elongated and fine-equiaxed grains. Red arrows in (**b**) refer to the region of the melt pool boundary.

**Figure 2 materials-16-00944-f002:**
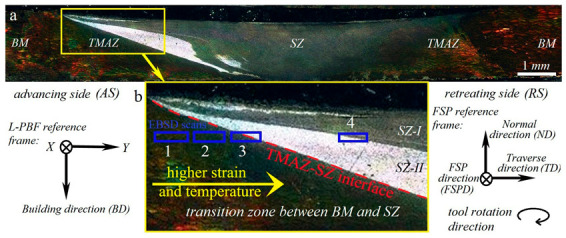
(**a**) Cross-sectional macrostructure of the 1-pass friction stir processed sample. (**b**) The higher magnification of the transition zone between BM and SZ on the advancing side and corresponding details of the L-PBF reference frame, FSP reference frame, and tool rotation direction. The blue rectangles indicate the different areas of EBSD scans.

**Figure 3 materials-16-00944-f003:**
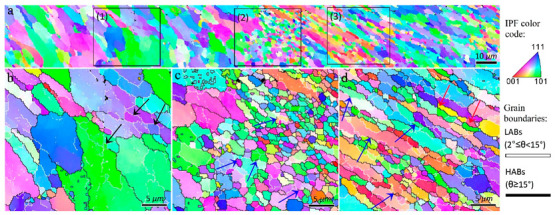
(**a**) Inverse pole figure maps of the rectangle 1 in Figure 2b. (**b**–**d**) Higher magnified views of the zones 1–3 indicated in (**a**). Black arrows in (**b**) refer to the LABs. Blue arrows in (**c**,**d**) show the continuous dynamic recrystallization mechanism occurring by transformation of LABs to HABs. The red arrows in (**d**) mention the fibrous grains in which the geometrically dynamic recrystallization (GDRX) arises.

**Figure 4 materials-16-00944-f004:**
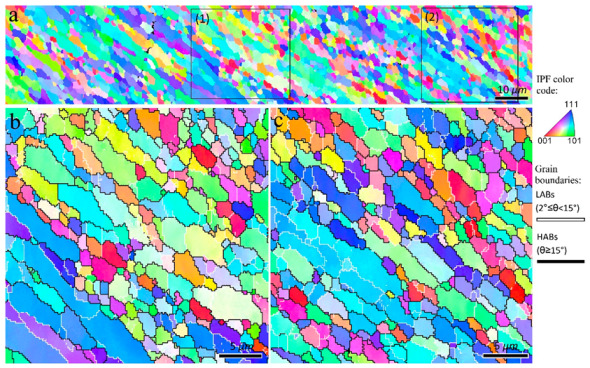
(**a**) Inverse pole figure maps of the TMAZ indicated by rectangle 2 in Figure 2b. Higher magnification of zones 1 and 2 indicated in (**a**) are illustrated in (**b**,**c**), respectively.

**Figure 5 materials-16-00944-f005:**
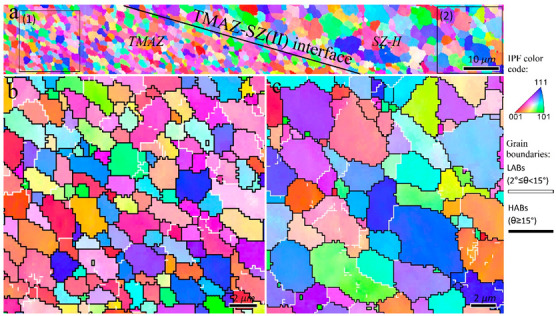
(**a**) Inverse pole figure maps of the interfacial zone between TMAZ and SZ-II regions in 1-pass sample indicated by rectangle 3 in Figure 2b. (**b**,**c**) Higher magnification of zones 1 and 2 indicated in (**a**), respectively.

**Figure 6 materials-16-00944-f006:**
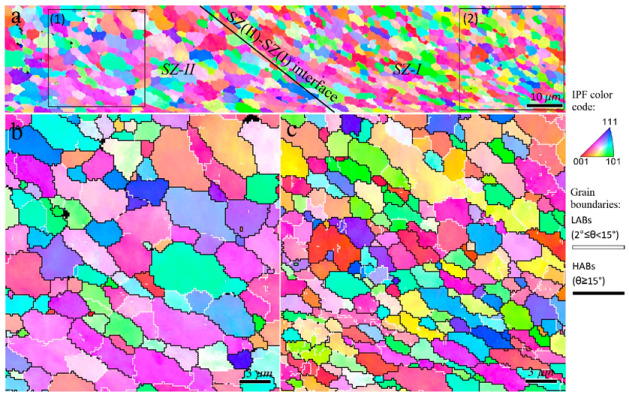
(**a**) Inverse pole figure maps of the interfacial zone between SZ-II and SZ-I regions in 1-pass sample indicated by rectangle 4 in Figure 2b. Higher magnification of zones 1 and 2 indicated in (**a**) are illustrated in (**b**,**c**), respectively.

**Figure 7 materials-16-00944-f007:**
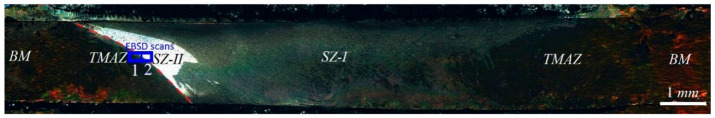
Cross-sectional macrostructure of 2-pass friction stir processed sample. The blue rectangles indicate areas 1 and 2 of EBSD scans. The details of the FSP reference frame are the same as Figure 2. The interface between TMAZ and SZ is depicted by a red-dashed line.

**Figure 8 materials-16-00944-f008:**
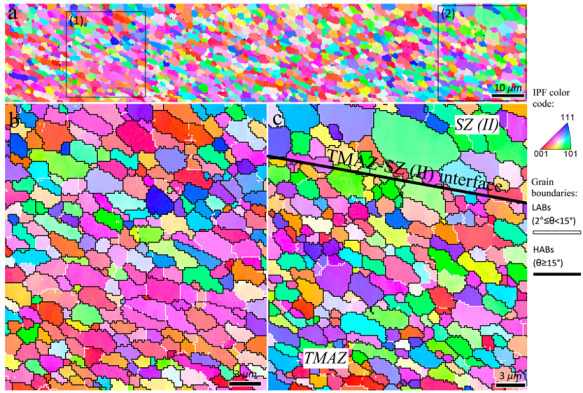
Inverse pole figure maps of the area indicated by rectangle 1 in Figure 7. Higher magnified views of zones 1 and 2 indicated in (**a**) are illustrated in (**b**) TMAZ and (**c**) interfacial area between TMAZ and SZ-II, respectively.

**Figure 9 materials-16-00944-f009:**
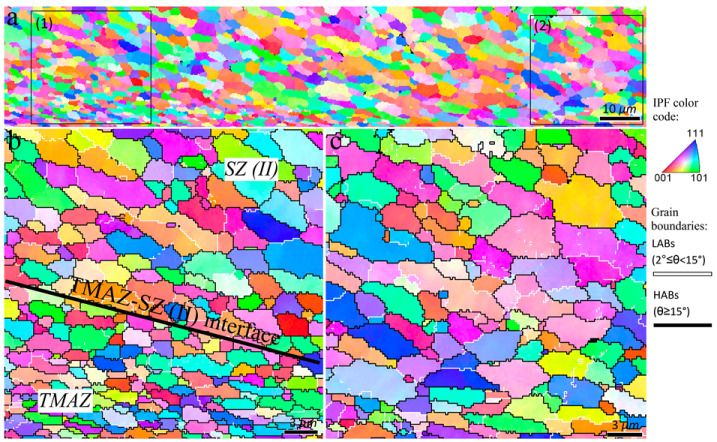
Inverse pole figure maps of the area indicated by rectangle 2 in Figure 7. Higher magnification of zones 1 and 2 indicated in (**a**) are illustrated in (**b**) interfacial area between TMAZ and SZ-II and (**c**) SZ-II, respectively.

**Figure 10 materials-16-00944-f010:**
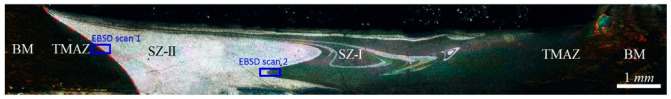
Cross-sectional macrostructure of 3-pass friction stir processed sample. The blue rectangles indicate the areas of EBSD scans. The details of the FSP reference frame are the same as Figure 2. Interface between TMAZ and SZ is depicted by a red-dashed line.

**Figure 11 materials-16-00944-f011:**
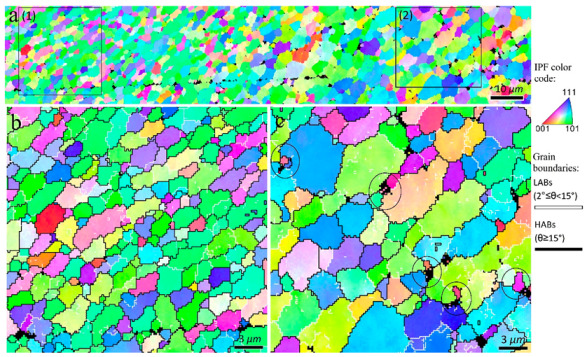
Inverse pole figure maps of the interfacial area between TMAZ and SZ-II in the 3-pass sample indicated by rectangle 1 in Figure 10. Higher magnification of zones 1 and 2 indicated in (**a**) are illustrated in (**b**,**c**), respectively. The circles in (**c**) refer to the non-index regions containing fine DRX grains, i.e., particle stimulated nucleation mechanism.

**Figure 12 materials-16-00944-f012:**
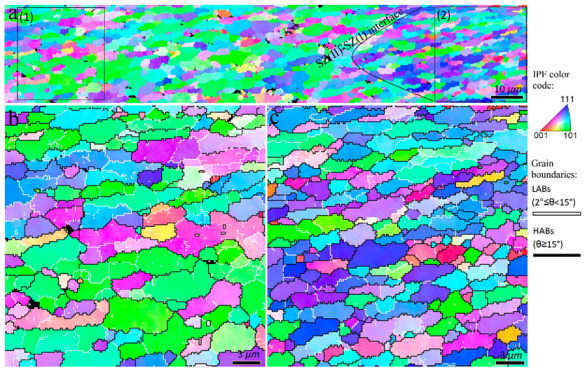
Inverse pole figure maps of the interfacial area between SZ-I and SZ-II in the 3-pass sample indicated by rectangle 2 in Figure 10). Higher magnification of zones 1 and 2 indicated in (**a**) are illustrated in (**b**,**c**), respectively.

**Figure 13 materials-16-00944-f013:**
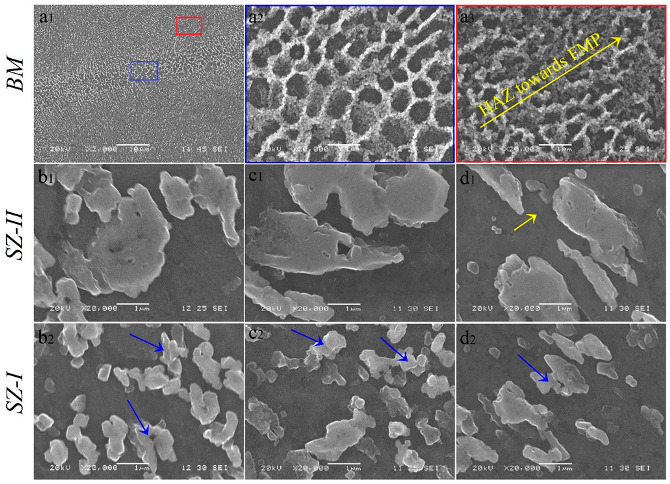
(**a**) SEM micrographs of as-built L-PBF AlSi10Mg at different magnification. The higher magnification of zones indicated by both blue and red rectangles in (**a1**) are shown in (**a2**,**a3**), respectively. (**b1**–**d1**) SEM images of SZ-I and (**b2**–**d2**) SZ-II in 1–3 pass FSPed samples. The yellow arrow in SZ-II of the 3-pass sample (**d1**) shows the grain boundary pinned by Si particles. Blue arrows in images related to SZ-I refer to the coalescence of Si particles during the growth.

**Figure 14 materials-16-00944-f014:**
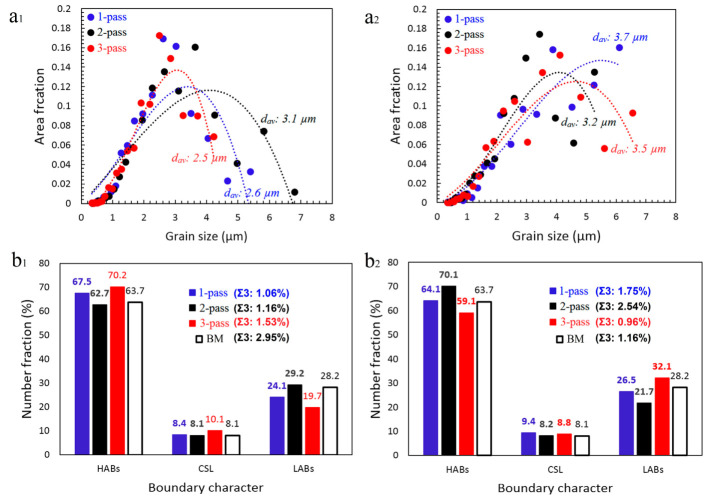
(**a**) Grain size and (**b**) grain boundary characterization distribution: (**a1**,**b1**) SZ-I and (**a2**,**b2**) SZ-II.

**Figure 15 materials-16-00944-f015:**
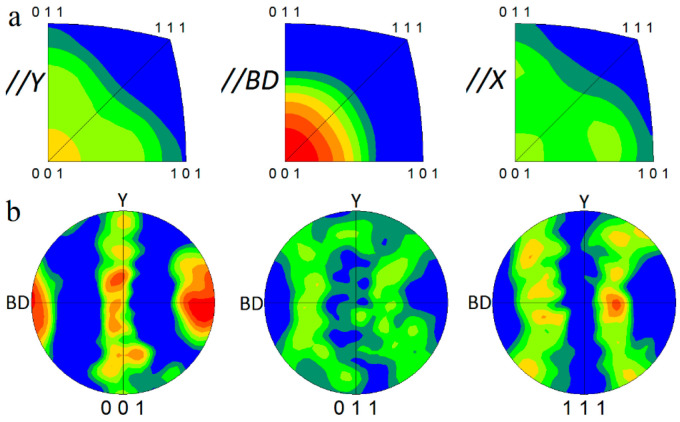
(**a**) Inverse pole figures parallel to x, y, and BD directions in conjunction with (**b**) (001), (011), and (111) pole figures (PFs) of as-built L-PBF AlSi10Mg.

**Figure 16 materials-16-00944-f016:**
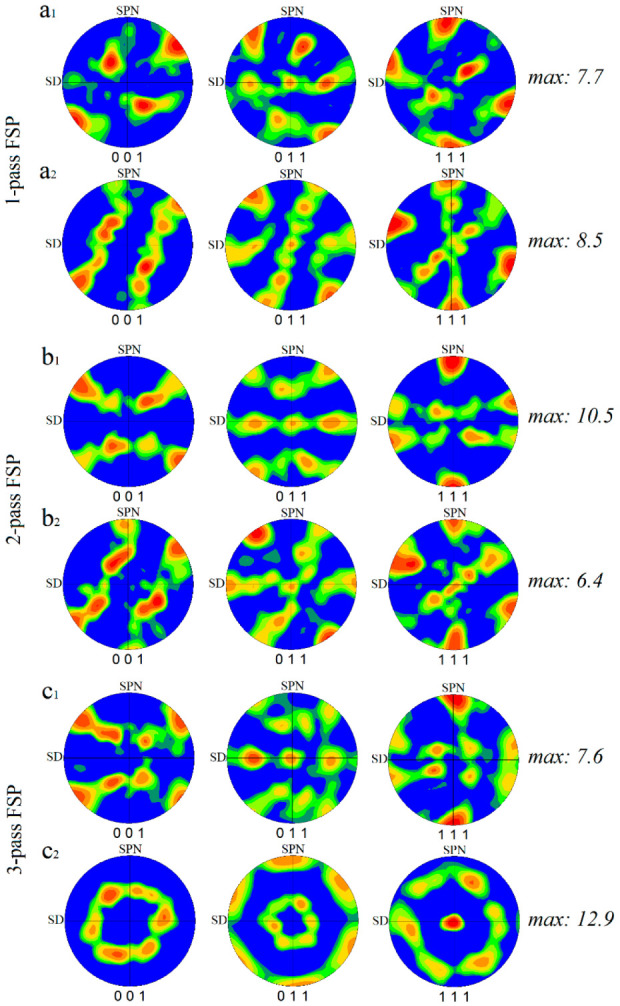
Rotated (001), (011), and (111) pole figures of SZ-I and SZ-II areas in (**a**) 1-pass, (**b**) 2-pass, and (**c**) 3-pass FSPed samples, respectively: (**a1**,**b1**,**c1**) SZ-I and (**a2**,**b2**,**c2**) SZ-II. The texture intensity values are written at the right-hand side of PFs.

**Figure 17 materials-16-00944-f017:**
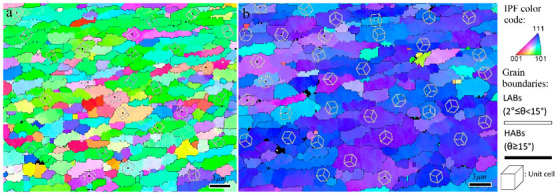
Inverse pole figure maps of SZ-I (**a**) and SZ-II (**b**) after applied rotations for texture analysis. The drawn cubes refer to the orientation of unit cells in individual grains.

**Table 1 materials-16-00944-t001:** Ideal texture components with corresponding Euler angles and miller indices in simple shear deformation of metals with a face centered cubic crystallographic structure [10,30].

Symbol	Euler Angles (^°^)			Miller Indices (hkl) <uvw>	(111) Pole Figure
	φ_1_	Φ	φ_2_		
$A_{1}^{*}$	35.26/215.26	45	0/90	$\left(111\right)\left[\bar{1}\bar{1}2\right]$	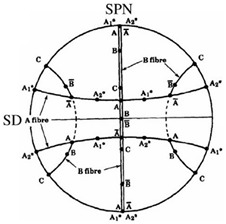
	125.26	90	45		
$A_{2}^{*}$	144.74	45	0/90	$\left(111\right)\left[11\bar{2}\right]$	
	54.74/234.74	90	45		
A	0	35.26	45	$\left(1\bar{1}1\right)\left[110\right]$	
$\bar{A}$	180	35.26	45	$\left(\bar{1}1\bar{1}\right)\left[\bar{1}\bar{1}0\right]$	
B	0/120/240	54.74	45	$\left(1\bar{1}2\right)\left[110\right]$	
$\bar{B}$	60/180	54.74	45	$\left(\bar{1}1\bar{2}\right)\left[\bar{1}\bar{1}0\right]$	
C	90/270	45	0/90	{001}〈110〉	
	0/180	90	45		

## Data Availability

The raw/processed data required to reproduce these findings cannot be shared at this time as the data also forms part of an ongoing study.

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
