# Peer review of "Grain Structure Formation and Texture Modification through Multi-Pass Friction Stir Processing in AlSi10Mg Alloy Produced by Laser Powder Bed Fusion"

_materials, 2023, doi:10.3390/ma16030944_

Round 1

Reviewer 1 Report

The authors systematically investigated the evolution of grain boundaries and crystallographic texture during the process of friction stir processing of the L-PBFed AlSi10Mg alloy. The study indicates that friction stir processing is a promising strategy to modify the structure manufactured by laser powder bed fusion. So, from my point of view, this manuscript can be accepted, but the following minor revisions need to be performed.

1.      The introduction can be appropriately extended. The content about how the parameters of friction stir welding or friction stir processing influence the microstructure and mechanical performance can be included in the introduction. For example, tool geometry, pin to shoulder ratio, shoulder surface features have been extensively investigated and summarized, and even the influence of the pin or tool eccentricity also has been studied. Some of the literatures are recommended and listed below:

https://doi.org/10.1007/s00170-022-09793-x; https://doi.org/10.1016/j.jallcom.2020.154045; https://doi.org/10.1016/j.matpr.2019.12.042

2.      Some description details should be revised. The authors stated that ‘Metallographic samples were cut from the joint perpendicular to the FSW direction’. It is not proper to use the term joint because it is a special term for welding. And the FSW direction should be revised into FSP direction.

3.      The format of the superscript and subscript should be revised, including mm3, 67â—¦, HNO3 and etc.

4.      The information (manufacturer name and model) of the FSP machine was not given.

5.      How was the plunge depth of 0.1 mm defined. Because a tilt angle of 2â—¦ was used, it is needed to illustrated the plunge depth using a schematic.

6.      The authors claimed that ‘the grains were elongated owing to the shear deformation induced by FSP, and the MPBs of the BM disappeared’. What is the evidence for that MPBs of the BM disappeared.

7.      For the explanation of the grain size difference between SZ-I and SZ-II, straining effect between the shoulder and pin can be considered.

8.      It is difficult to understand that the grain size of the TMAZ is smaller than that in SZ- II in Fig.8c and Fig.9b.

9.      ‘The corresponding microstructure is proposed to enhance the corrosion resistance of L-421 PBF AlSi10Mg’ can not be listed in the conclusion.

Author Response

My reply to the Reviewer is submitted.  

Reviewer 2 Report

The paper can be published in present form.

Author Response

My reply to the Reviewer is submitted.

Reviewer 3 Report

The manuscript submitted for the review is entitled: Grain boundary engineering and texture modification through multi-pass friction stir processing in AlSi10Mg alloy produced by laser powder bed fusion.

I like the idea presented in the title, as indeed there are potential problems with L-PBF metal parts due to very directional nature of the 3D printing process. FSP can be applied iteratively to introduce changes in microstructure and texture, while the heat induced during the processing drives the recovery processes. Therefore concept of using multi pass FSP for GBE seems both valid and interesting.

However in this particular case I find the title of the manuscript to be misleading.

Out of 400 lines (excluding the abstract and references) and 17 figures there are actually only 5 lines and one figure in the discussion section that actually mention GBE concepts, that is the idea of grain boundary character distribution and increase in CSL boundaries quantity:

“From Figs. 14b and c, multi-pass FSP generated more CSL boundaries of 14% compared to 1 and 2-pss samples. The number of low energy CSL boundaries, i.e. CSLs with Σ≤27, increased by up to 9.5% and 11.6% in SZ-I and SZ-II in 324 the 3-pass FSPed sample (Figs. 14c1 and c2). Moreover, the Σ3 boundaries have a considerable fraction in CSLs, equal to 1.7% and 3.4% for SZ-I and SZ-II, respectively.”

Moreover, the presented CSL result is questionable, as there is no reference value for CSL number fraction in the base material (before FSP), therefore it is hard to judge the actual impact of the FSP on the GBCD.

In addition the reported increase of CSL frequency 14% after 3 FSP passes in comparison to 13% after 1 and 2 passes might be statistically unsignificant, especially as the results are seemingly based on only 2 EBSD maps with relatively limited grain statistics, and CSL boundaries with sigma=>29, which are normally considered as random, are also included in the total CSL count.

There is also no mention of the criterion used for CSL boundary classification, so it is impossible to judge how accurate the results are.

The second part of the title mentions the texture modification, which is represented in more detail in the actual manuscript, however the texture analysis seems to be rather qualitative. Authors mention the appearance of various texture components based on the presented pole figures and attribute them to deformation modes and dynamic recovery mechanisms active during FSP.

The majority of the manuscript, contrary to the manuscript title is devoted to the analysis of microstructure evolution and identification of grain refinement mechanisms after consecutive FSP passes. Authors point to different recovery processes responsible for microstructure evolution in various areas of the FSPed material, such as dynamic recovery, continuous dynamic recrystallization and geometric dynamic recrystallization, which changes based on the proximity to the actual stir zone. Authors also point to the particle stimulated nucleation, and grain boundary impingement as a source of microstructure refinement.

I don’t have major remarks considering this section of the manuscript apart from couple mentioned below. However it is difficult for me to judge the novelty of presented results as authors do not point out what new information on the microstructure behavior or the relationship between the FSP parameters and the microstructure is obtained from this study.

Overall the quality of presented results is relatively good, however as mentioned above, there is no clear novelty immediately apparent from reading of the manuscript, and authors also don’t seem to highlight one either. The GBE portion is lacking in my opinion. The microstructure and texture evolution is documented reasonably well, however the description is mostly qualitative, and therefore it is difficult to draw conclusions on the relation of the FSP parameters and the actual microstructure changes.

The writing of the manuscript is acceptable for the most parts, however the overuse of abbreviations, symbols and excessive citing of numerical results in text makes the manuscript quite hard to read in some parts.

Some other remarks:

140-141 The formation of subgrains surrounded by LABs (black arrows in Fig. 3b) indicates the occurrence of dynamic recovery (DRV) in this zone.

-        Subgrains and LABs are also present in the base material, therefore conclusion with DRV might not be sufficiently justified

194-195: The correlation between Fig. 6 and Eqs. 194 1 and 2 indicate that (…)

-        There is no such thing as correlation between a figure and a equation. Also the idea behind this sentence unclear. Authors suggest that in one part of the material the microstructure evolution is driven by temperature and in other by strain rate, however this should be explained more plainly and also since the EBSD is used for microstructure analysis, this can be represented quantitatively.

 246-249 However, from the viewpoint of grain orientation, the third pass led to the formation of a strong crystallographic texture due to the rotation of the DRX grains. The existence of con- siderable LABs inside the grains confirmed the dynamic nature of DRX, which was due to their post-deformation during FSP.

-        First sentence refers to different texture forming in 3 pass, suggesting that texture was previously discussed for passes 1 and 2, but this is the first mention of the texture in the results section of the manuscript

-        The second sentence is written very poorly, its unclear what is the subject, the second part of the sentence is supposed to point to.

409-410 The bimodal grain structure results in the formation of two distinct zones: SZ-I with a fine grain size and SZ-II with a coarse grain size in the SZ.

-        Should be: “structure results from

Authors seem to be using word “frame” instead of the “reference frame”, and those are not the same thing.

In my opinion the manuscript should be improved and resubmitted before further consideration for publication.

As a paper focused on GBE and texture modification, the analysis should be much more robust, and also the demonstration of the introduced modifications on relevant properties would increase the value of manuscript.

As a paper focused on microstructure modification, more quantitative analysis, discussion on the relationship between processing parameters, microstructure and properties is needed, as well as clear statement of prime novelty should be expected.

Author Response

My reply to the Reviewer is submitted.

Round 2

Reviewer 3 Report

1.       Changes made by authors to the title, abstract and introduction, in response to reviewer’s comments are in the right direction and increase the readability and accessibility of the manuscript for the readers. Updated title is now accurately reflecting the actual contents of the manuscript.

2.       In response to my comment:

194-195: The correlation between Fig. 6 and Eqs. 194 1 and 2 indicate that (…)

-          There is no such thing as correlation between a figure and a equation. Also the idea behind this sentence unclear. Authors suggest that in one part of the material the microstructure evolution is driven by temperature and in other by strain rate, however this should be explained more plainly and also since the EBSD is used for microstructure analysis, this can be represented quantitatively.

The authors have introduced some changes to the cited section (lines 210-214 in revised manuscript)

In my opinion the revised version of this section is still lacking and also again written poorly.

In my opinion if the authors are interested in using ZH parameter to analyze the microstructure results, the following questions should be considered and addressed in the manuscript:

Does the qualitative analysis using ZH parameter and the eqs 1 and 2 explain existence of two separate zones (SZ-I, SZ-II) with different grain size, and sharp interface between those zones?

Or should a gradual change of the microstructure be expected if the eqs 1 and 2 can be used to describe the relation between deformation parameters and resulting microstructure during FSP?

Is there a combination of temperature and strain rate gradients possible to achieve during FSP that when applied to eqs 1 and 2 would result in average grain size obtained in the EBSD measurements for SZ-I and SZ-II?

3.       In my opinion the manuscript still requires a careful proof-read and editing, preferably by a native English speaker in order to improve the language and overall readability of the manuscript.

4.       In response to my original comment:

The writing of the manuscript is acceptable for the most parts, however the overuse of abbreviations, symbols and excessive citing of numerical results in text makes the manuscript quite hard to read in some parts.

The authors responded by adding a abbreviation/definition list at the end of manuscript

The problem of abbreviation overuse in numerous sentences is not solved by attaching the list of abbreviations/symbols, and in my opinion such list should not be included in the final publication.

The problems mentioned in my original comment can only be solved by rewriting selected parts of the manuscript using more plain and descriptive language pointing out observed qualities and emerging relations.

Therefore I do not believe the authors addressed my concerns from the original comment.

However since this problem refers to the editing aspect of the manuscript, rather than scientific one, the decision if the manuscript should be accepted in current form is in the hands of the editor.

My recommendation is that the manuscript can only be published if the authors sufficiently address points 2 and 3 mentioned above. It would be advisable to also sufficiently address point 4. However I leave the decision with the editor in this case.

Author Response

The response is attached as a word file.
